# Sex Differences in the Expression of Cardiac Remodeling and Inflammatory Cytokines in Patients with Obstructive Sleep Apnea and Atrial Fibrillation

**DOI:** 10.3390/biomedicines12061160

**Published:** 2024-05-23

**Authors:** Chun-Ting Shih, Hui-Ting Wang, Yung-Che Chen, Ya-Ting Chang, Pei-Ting Lin, Po-Yuan Hsu, Meng-Chih Lin, Yung-Lung Chen

**Affiliations:** 1Division of Cardiology, Department of Internal Medicine, Kaohsiung Chang Gung Memorial Hospital, Chang Gung University College of Medicine, Kaohsiung 833, Taiwan; phage0925@cgmh.org.tw (C.-T.S.); r40391132@gmail.com (P.-T.L.); 2Emergency Department, Kaohsiung Chang Gung Memorial Hospital, Kaohsiung 833, Taiwan; gardinea1983@gmail.com; 3School of Medicine, College of Medicine, National Sun Yat-sen University, Kaohsiung 804, Taiwan; 4Division of Pulmonary & Critical Care Medicine, Department of Internal Medicine, Kaohsiung Chang Gung Memorial Hospital, Chang Gung University College of Medicine, Kaohsiung 833, Taiwan; yungchechen@yahoo.com.tw (Y.-C.C.); hsupowan@yahoo.com.tw (P.-Y.H.); 5Department of Neurology, Kaohsiung Chang Gung Memorial Hospital, Chang Gung University College of Medicine, Kaohsiung 833, Taiwan; emily0606@cgmh.org.tw; 6Graduate Institute of Clinical Medical Sciences, College of Medicine, Chang Gung University, Taoyuan 333, Taiwan

**Keywords:** atrial fibrillation, apnea–hypopnea index, cardiac remodeling, *GJA1*, inflammatory cytokines, obstructive sleep apnea, sex-specific differences

## Abstract

Although there is a link between obstructive sleep apnea (OSA) and atrial fibrillation (AF) and numerous investigations have examined the mechanism of AF development in OSA patients, which includes cardiac remodeling, inflammation, and gap junction-related conduction disorder, there is limited information regarding the differences between the sexes. This study analyzes the impact of sex differences on the expression of cardiac remodeling, inflammatory cytokines, and gap junctions in patients with OSA and AF. A total of 154 individuals diagnosed with sleep-related breathing disorders (SRBDs) were enrolled in the study and underwent polysomnography and echocardiography. Significant OSA was defined as an apnea–hypopnea index (AHI) of ≥15 per hour. Exosomes were purified from the plasma of all SRBD patients and incubated in HL-1 cells to investigate their effects on inflammatory cytokines and *GJA1* expression. The differences in cardiac remodeling and expression of these biomarkers in both sexes were analyzed. Of the 154 enrolled patients, 110 patients were male and 44 patients were female. The LA sizes and E/e’ ratios of male OSA patients with concomitant AF were greater than those of control participants and those without AF (all *p* < 0.05). Meanwhile, female OSA patients with AF had a lower left ventricular ejection fraction than those OSA patients without AF and control subjects (*p* < 0.05). Regarding the expression of inflammatory cytokines and *GJA1*, the mRNA expression levels of *GJA1* were lower and those of *IL-1β* were higher in those male OSA patients with AF than in those male OSA patients without AF and control subjects (*p* < 0.05). By contrast, mRNA expression levels of *HIF-1α* were higher in those female OSA patients with and without AF than in control subjects (*p* < 0.05). In conclusion, our study revealed sex-specific differences in the risk factors and biomarkers associated with AF development in patients with OSA.

## 1. Introduction

Obstructive sleep apnea (OSA) is a widely prevalent medical condition characterized by intermittent oxygen deprivation caused by recurrent upper airway obstruction during sleep [1]. Atrial fibrillation (AF), a common cardiac arrhythmia, has a significant global prevalence (2% to 4%) and is associated with heightened risks of stroke, heart failure (HF), cognitive impairment, depression, hospitalization, and mortality [2]. OSA is considered a potential risk factor for AF, implicated through mechanisms such as inflammatory responses, autonomic dysfunction, and negative thoracic pressure contributing to cardiac remodeling [3,4,5,6,7,8].

Sex differences in OSA and AF prevalence, clinical presentation, comorbidities, and pathophysiological mechanisms have been noted [9,10]. The prevalence of OSA is 27.3% in males, with a range of 9% to 86%, and 22.5% in females, with a range of 3.7% to 63.7%. Females often exhibit lower apnea–hypopnea index (AHI) scores but higher rates of symptomatic presentation, hypertension, and depression. Conversely, males show higher rates of diabetes and ischemic heart disease. Notably, HF with preserved ejection fraction (HFpEF) has a higher prevalence among females, while HF with reduced ejection fraction (HFrEF) is more prevalent in males [9,11].

AF incidence is higher in men, with female AF patients tending to be older and having higher rates of hypertension, valvular heart disease, and HFpEF but lower rates of coronary artery disease. Women with AF experience more severe symptoms and may have poorer responses to antiarrhythmic therapy or catheter ablation [12,13]. The pathophysiology of AF involves electrical and structural remodeling as well as autonomic neural regulation, with potential sex-specific differences [14,15].

Our study aimed to explore sex differences in individuals with concurrent OSA and AF, focusing on comorbidities, clinical presentation, cardiac remodeling, and pathophysiological mechanisms, as limited evidence currently exists in this area.

## 2. Materials and Methods

### 2.1. Patient Enrollment and Sample Management

This study enrolled individuals with sleep-related breathing disorders (SRBDs) in Kaohsiung Chang Gung Memorial Hospital between May 2019 and November 2022. Patients previously diagnosed and treated for OSA were excluded from the study. Other exclusions were made for individuals with acute infections, autoimmune disorders, malignancies, or those under 30. To mitigate confounding factors from severe systolic HF causing central sleep apnea and AF, patients with HFrEF were excluded from the analysis.

After enrollment, participants promptly underwent polysomnography (PSG), peripheral blood (PB) sampling, and echocardiography. In this investigation, patients did not undergo CPAP therapy or AF ablation until they completed all evaluations. Plasma derived from PB was utilized for exosome quantification and purification. Clinical features, including age, sex, body mass index (BMI), comorbidities, Epworth Sleepiness Scale (ESS) scores, PSG data, and echocardiographic parameters such as the dimensions of the aorta and the left atrium (LA), the thickness of the interventricular septum (IVS) and the left ventricular posterior wall (LVPW), the left ventricular end-diastolic and end-systolic volumes (LVEDV and LVESV), and left ventricular ejection fraction (LVEF) and E/e’ ratio were evaluated.

Before enrolling in the research, all participants were required to provide written informed consent. The research was approved by an institutional review board at Chang Gung Memorial Hospital (IRB numbers: 201801943B0 and 201801943B0C501~4) and complied with the tenets of the Declaration of Helsinki.

### 2.2. Study of Overnight PSG and Determination of SRBD Metrics

All study participants were subjected to PSG investigations overnight. Our prior studies provided in-depth descriptions of PSG methods [16,17,18]. The nightly PSG investigation was conducted at our hospital’s sleep department using a commercially available standardized suite (Sandman Elite, Mallinckrodt Inc., St. Louis, MO, USA). Skilled technicians identified, analyzed, and recorded sleep parameters according to predetermined criteria [19].

Apnea was characterized as the absence of nasal airflow for a minimum duration of 10 s, while hypopnea was operationally defined as a nasal airflow reduction of 30% or greater that persisted for a minimum of 10 s in conjunction with an arousal event or a decrease in pulse oximeter oxygen saturation surpassing 4% from the initial value. Significant OSA was operationally defined as an AHI equal to or greater than 15 events per hour accompanied by excessive daytime somnolence and sleep apneas. Patients who had an AHI of less than 15 per hour were deemed the control group in this study. The oxygen desaturation index (ODI) was determined by calculating the mean number of desaturation events per hour of sleep. Desaturation episodes were characterized by a reduction in oxygen saturation levels by 4% or more relative to the baseline value established before the event, with a minimum duration of 10 s.

### 2.3. Definition of AF, HF with Preserved or Mildly Reduced EF, and Echocardiographic Parameters

Using the 12-lead electrocardiogram (ECG) or 24 h ambulatory ECG, all study participants were evaluated, and clinical diagnoses of AF were made following the ACC/AHA/ESC 2016 criteria [20]. AF was characterized by the absence of *p* waves on an ECG and their replacement with fibrillatory waves. HFpEF or HF with a mildly reduced ejection fraction (HFmrEF) was defined as the presence of symptoms and/or signs of HF, an LVEF ≥ 50% (HFpEF) or 41–49% (HFmrEF), and objective evidence of cardiac anatomical and/or functional abnormalities that indicate the presence of left ventricular diastolic dysfunction or elevated left ventricular filling pressures, according to the 2021 ESC Guidelines for the diagnosis and treatment of acute and chronic HF [21].

Adhering to our established echocardiography protocol, we used a Sonos 7500 device (Live 3D Echo technology; Philips Medical Systems) to assess echocardiographic data [22,23,24]. The dimensions of the aorta, the LA, and the thickness of the IVS and the LVPW were measured. Additionally, LVEDV, LVESV, LVEF, and E/e’ were determined. The aorta’s diameter was assessed using M-mode echocardiography in the parasternal long-axis view, typically at the aortic root. LA size was assessed using two-dimensional echocardiography, measuring the anteroposterior and transverse diameters in the parasternal long-axis view. Using M-mode echocardiography with the parasternal long-axis view, the interventricular septum and the posterior wall of the left ventricle were assessed for thickness. Volume measurements were obtained using Simpson’s method, tracing the endocardial borders of the left ventricle in both end-diastolic and end-systolic phases from multiple echocardiographic views. LVEF was calculated using Teichholz’s or Simpson’s biplane method, assessing changes in left ventricular volume during the cardiac cycle. The E/e’, representing the ratio of E to the mean of the septal and lateral e’ velocities, was calculated by assessing transmitral Doppler (E wave) and tissue Doppler (e’ wave) velocities. As the peak velocity of early mitral inflow (E) was documented, tissue Doppler imaging facilitated the measurement of the early peak diastolic velocity (e’) at the septal and lateral mitral annulus. In patients with AF, the E/e’ ratio was determined according to our hospital’s standard echocardiography procedure, typically using a single representative beat assessment or index beat assessment, where E and e’ were selected to reflect an average value, as judged by an experienced cardiologist.

### 2.4. HL-1 Cell Culture

The HL-1 Cardiac Muscle Cell Line’s pooled primary cells were procured from SIGMA (SIGMA-ALDRICH Corp., St. Louis, MO, USA) and cultivated in a blend of Claycomb Basal Medium, following an established methodology [25]. HL-1 cells were meticulously seeded in a 6-well plate at a density of 4 × 10^5^ cells/well.

### 2.5. Isolation, Purification, Quantification, and Incubation of Exosomes with HL-1 Cells

The process of isolating, purifying, and quantifying exosomes followed established methodologies [16,26,27,28]. PB samples were collected from individuals diagnosed with SRBDs. Subsequently, HL-1 cells underwent a 24 h treatment at 37 °C with exosomes obtained from patients diagnosed with both significant and non-significant OSA at a concentration of 10 µg/mL.

### 2.6. Real-Time Quantitative Reverse Transcriptase–Polymerase Chain Reaction Analysis (qRT-PCR) of mRNA Expression in HL-1 Cells

Following the aforementioned protocol, HL-1 cells were isolated, and RNA was extracted. Subsequently, complementary DNA (cDNA) was synthesized, and qRT-PCR analysis was performed [29,30]. The mRNA expression levels of inflammation-associated cytokines [*Tumor necrosis factor (TNF)-α*, *Interleukin (IL)-1β*, *IL-6*, and *Transforming growth factor (TGF)-β*], *Hypoxia-inducible factor (HIF)-1α*, and *Gap junction alpha-1 (GJA1)* were normalized using the internal control *GAPDH* to calculate the relative threshold cycle (ΔCt). The experimental procedure was duplicated to ensure the replication of each reaction.

### 2.7. Statistical Analysis

Our recent study found that OSA patients with AF had an LA size of 40.3 ± 6.4 mm. In contrast, OSA patients without AF had an LA size of 35.7 ± 4.6 mm [16]. The patient number ratio was calculated to be 1:4 when comparing individuals with and without AF. Based on a one-sided type I error rate of 0.025 and a power of 0.9, it was determined that a minimum of 100 OSA patients needed to be enrolled in the study. Out of these, 20 OSA patients had AF, and 80 did not have AF. Moreover, according to previous studies, the ratio of females to males was 1:4 [31,32]. The projected minimum number of female patients diagnosed with OSA and AF was 4, whereas the minimum number of male patients was 16.

All patients and subgroups are given detailed summaries. Quantitative data are displayed as percentages, means ± standard deviations, or medians with interquartile ranges. The Kruskal–Wallis H test was used to assess relationships between categorical variables in individuals with significant OSA and AF, those with significant OSA without AF, and those without significant OSA. For continuous data, appropriate statistical procedures, such as the Student’s *t*-test, the Mann–Whitney U test, and one-way ANOVA, were employed. The 2^−ΔΔCt^ method was used to quantify relative changes in mRNA gene expression levels in HL-1 cells. All statistical analyses were performed using SPSS version 17.0 (SPSS, Chicago, IL, USA).

## 3. Results

### 3.1. Baseline Characteristics and Echocardiographic Data of Those OSA Patients with and without AF and Control Subjects

Between May 2019 and November 2022, we prospectively enrolled 154 patients with SRBDs. Among them, 110 (71.4%) were males, with an average age of 55 ± 11 years. One hundred and twenty-six (81.8%) had significant OSA with an AHI ≥ 15 per hour, and 28 (18.2%) did not. A summary of the distinctions between the OSA patients with and without AF and the control subjects is provided in Table 1. In brief, significant OSA was associated with an increased BMI, AHI, ODI, and duration of oxygen saturation (SpO_2_) < 90%, in addition to lowest and mean SpO_2_, compared to patients without OSA (all *p* < 0.05). The OSA patients with concomitant AF were older, a larger proportion of them were male, and there were more OSA patients with HF than OSA patients without AF and control subjects (all *p* < 0.05). Usage of medications, including beta-blockers, calcium channel blockers, antiarrhythmic drugs, and direct oral anticoagulants for AF treatment, was higher in patients with OSA and AF compared to OSA patients without AF and control subjects. Patients with and without clinically significant OSA exhibited comparable baseline characteristics, including sex, alcohol use, diabetes mellitus (DM), hypertension, HF, stroke, coronary artery disease, vital signs at enrollment, and ESS score. Regarding echocardiographic evaluation, OSA patients with comorbid AF had a larger LA size and E/e’ and a lower LVEF compared with OSA patients without AF and control subjects (all *p* < 0.05) (Table 1).

### 3.2. mRNA Gene Expression of HIF-1α and Inflammatory Cytokines in HL-1 Cells Treated with Exosomes Obtained from OSA Patients with and without AF and Control Subjects 

To determine the mechanism and effect of intermittent hypoxemia events on atrial cardiomyocytes, exosomes obtained from OSA patients with and without AF, as well as control subjects, were used to treat HL-1 cells. The mRNA expression level of *GJA1* was lower in those OSA patients with AF than in those without AF and control subjects (*p* < 0.05). OSA patients with AF also had higher mRNA expression levels of *IL-1β* than OSA patients without AF and higher mRNA expression levels of *HIF-1α* and *TNF-α* than control subjects (all *p* < 0.05). In addition, OSA patients without AF had higher *HIF-1α* expression than control subjects (*p* < 0.05). There was no difference in the mRNA expression of *IL-6* and *TGF-β* among these three groups (Figure 1).

### 3.3. The Differences in Baseline Characteristics and Echocardiographic Data among Those Male OSA Patients with and without AF and Control Subjects

The differences among male OSA patients with and without AF and control subjects are summarized in Table 2. Male patients with significant OSA exhibited higher BMI, AHI, and ODI and lower lowest and mean SpO_2_ values (all *p* < 0.05). Those patients with coincident AF were older and included more patients with HF than male OSA patients without AF and control subjects (both *p* < 0.05). No differences were observed among the three groups in other baseline characteristics, except for the usage of medications for AF treatment, including beta-blockers, calcium channel blockers, antiarrhythmic drugs, and direct oral anticoagulants, which was higher in patients with OSA and AF compared to OSA patients without AF and control subjects. Male OSA patients with coincident AF had larger LA sizes and E/e’ ratios than those without AF and control subjects (all *p* < 0.05). Furthermore, male OSA patients with AF had lower LVEFs than control subjects (*p* < 0.05). 

### 3.4. mRNA Gene Expression of HIF-1α and Inflammatory Cytokines in HL-1 Cells Treated with Exosomes Obtained from Those Male OSAS Patients with and without AF and Control Subjects

To assess the impact of intermittent hypoxemia episodes on atrial cardiomyocytes and the underlying mechanism, exosomes obtained from male patients with OSA and control subjects were utilized to treat HL-1 cells. The mRNA expression levels of *GJA1* were lower and those of *IL-1β* were higher in the male OSA patients with AF than in the male OSA patients without AF and the control subjects (*p* < 0.05). Male OSA patients with AF had higher mRNA expression levels of *TNF-α* than control subjects (*p* < 0.05). The mRNA expression levels of HIF-1α, TGF-β, and IL-6 did not differ among the three groups (Figure 2).

### 3.5. The Differences in Baseline Characteristics and Echocardiographic Data among Those Female OSA Patients with and without AF and Control Subjects

The differences among female OSA patients with and without AF and control subjects are summarized in Table 3. Female patients with significant OSA had a higher AHI and ODI (both *p* < 0.05). Female patients with OSA and AF had a lower lowest SpO_2_ than the control subjects. There were no differences among the three groups in the other baseline characteristics. Female OSA patients with AF had a lower LVEF than the OSA patients without AF and the control subjects (*p* < 0.05). Additionally, female OSA patients with AF had a larger LA size than control subjects (*p* < 0.05).

### 3.6. mRNA Gene Expression of HIF-1α and Inflammatory Cytokines in HL-1 Cells Treated with Exosomes Obtained from Female Significant OSAS Patients with and without AF and Control Subjects

To assess the impact of intermittent hypoxemia episodes on atrial cardiomyocytes and the underlying mechanism, exosomes obtained from female patients with OSA and control subjects were utilized to treat HL-1 cells. The mRNA expression levels of *HIF-1α* were higher in the female OSA patients with and without AF than in the control subjects (*p* < 0.05). There was no difference in the mRNA expression of *TNF-α*, *IL-1β*, *IL-6*, *TGF-β*, and *GJA1* among these three groups (Figure 3).

### 3.7. The Relationship between Age, BMI, Echocardiographic Parameters (LA Size, LVEF, and E/e’ Ratio), Gene Expression of Biomarkers (IL-1β and GJA1), and AF Occurrence in OSA Patients

A correlation analysis was performed to evaluate the relationship between age, BMI, and the results, including echocardiographic parameters (LA size, LVEF, and E/e’ ratio) and gene expression of biomarkers (*IL-1β* and *GJA1*), in OSA patients. The results showed no significant association between age, BMI, and these parameters (Appendix A). The association between these parameters and AF occurrence in OSA patients after adjusting age and BMI was also analyzed. The results revealed that age and BMI did not significantly modulate the result (Appendix A).

## 4. Discussion

This study yielded several noteworthy findings. Firstly, individuals with OSA and AF were observed to be older and to have a higher prevalence of HF with preserved or mildly reduced EF, larger LA sizes, lower LVEF, and a higher E/e’ compared to those with OSA but without AF. Secondly, distinct sex-related variations were identified among individuals with both OSA and AF. Male OSA patients with AF were older and had a higher prevalence of HF with preserved or mildly reduced EF, larger LA sizes, and a higher E/e’ compared to their counterparts without AF. Conversely, female OSA patients with AF exhibited a notable decrease in LVEF compared to females without AF. Thirdly, concerning the potential mechanism underlying concurrent AF development in OSA patients, our study revealed that OSA patients with AF had lower expression of *GJA1* compared to OSA patients without AF and control subjects. Additionally, OSA patients with AF demonstrated higher expression of *IL-1β* compared to OSA patients without AF and higher expression of *TNF-α* and *HIF-1α* compared to controls. Regarding the distinct expressions in male and female OSA patients, males with OSA and AF exhibited higher *IL-1β* and lower *GJA1* expression than those without AF. Moreover, males with OSA and AF tended to have higher *TNF-α* and *IL-1β* expression and lower *GJA1* expression than control subjects. Furthermore, females, irrespective of AF status, displayed higher expression of *HIF-1α* than control subjects, suggesting the potential impact of hypoxia on OSA development. However, no difference in *HIF-1α* expression was observed between female OSA patients with and without AF.

Previous studies have identified elderly individuals and those with comorbidities such as HF, DM, renal failure, and reduced diastolic function as predictors of incident AF in OSA patients [31,33,34]. Our study further corroborated these findings, highlighting that advanced age, HFpEF or HFmrEF, increased LA size, elevated E/e’, and decreased LVEF are associated with a heightened risk of AF in OSA patients. Increased E/e’ and LA size in our study signified augmented LA pressure and diastolic dysfunction [35], diastolic dysfunction, LA size, and LVEF having been established as crucial predictors of AF development in diverse cohort studies [16,36,37,38]. Our findings validate prior research by confirming that both age and a history of HFpEF or HFmrEF are predominant risk factors for AF in OSA patients. Notably, our study revealed that increased LA size and E/e’ pose a higher risk of AF development, especially in male OSA patients. Furthermore, a lower LVEF emerged as a risk factor for AF development in both males and females.

Several studies have shown that OSA has the potential to induce cardiac remodeling and systemic inflammation, which may contribute to the initiation and maintenance of AF [17,39,40,41,42,43]. Evidence is accumulating in the literature that the *GJA1* gene is a crucial susceptibility factor for AF. A well-established AF locus, chromosome 6q22.31, houses the *GJA1* gene [44]. Recent studies suggested that the single-nucleotide polymorphism (SNP) rs13216675, situated near the *GJA1* gene on chromosome 6q22, was significantly associated with the risk of AF [45]. The *GJA1* gene may be associated with abnormal atrial automaticity and involved in both the onset and persistence of AF [46]. Our study revealed a decreased expression of *GJA1* in patients with co-existing AF and OSA compared to those OSA patients without AF. Connexin-43 (Cx43), a well-known connexin protein that affects cardiac conduction, is encoded by the *GJA1* gene [47,48]. A previous study showed that Cx43 downregulation contributed to sympathetic AF by affecting intercellular channel conduction [46]. OSA-associated increased sympathetic tone, resulting in decreased expression of Cx43, may contribute to AF occurrence in OSA patients. Cx43 expression modulation could represent an innovative therapeutic strategy for AF. Additionally, our previous study demonstrated that the AHI and ODI in the non-rapid eye movement period (when the autonomic balance shifts from sympathetic to parasympathetic dominance) of OSA patients with AF exhibited a negative correlation with HL-1 cell *GJA1* expression. This supports the notion that *GJA1* likely has a significant impact on the progression of AF in individuals with OSA [16]. Interestingly, our study revealed the lowest *GJA1* expression in OSA patients with AF and the highest expression in patients without OSA and AF in men. Still, there was no difference among the three groups in women. The assessment of heart rate variability indicated that women, compared to men of similar age, exhibited a dominance of vagal activity [49]. A recent study revealed that male patients with OSA exhibited higher LF/HF ratios compared to their female counterparts [50]. Further research should investigate the influence of autonomic dysfunction on the expression of *GJA1* and the development of AF in OSA patients of different sexes. 

Our investigation revealed higher levels of *TNF-α* and *IL-1β* expression in patients with both AF and OSA compared to individuals without OSA and AF. This difference was particularly significant in men but not in women. The development of AF, associated with pro-fibrotic processes, is linked to the inflammatory cascade. Well-documented biomarkers, such as *TNF-α*, *IL-6*, and *IL-1β*, have consistently been observed in both animal models and individuals affected by OSA [51,52]. Our previous study also demonstrated increased expression of *HIF-1α*, *TNF-α*, *IL-6*, and *TGF-β* in the intermittent hypoxia/reoxygenation condition [18]. Accumulating evidence supports the pivotal role of *TNF-α* and *IL-1β* in atrial remodeling, including structural, electrical, contractile, and autonomic remodeling, which may play a crucial role in the pathogenesis of AF. Moreover, prior research has indicated that *TNF-α* can influence the function of the sympathetic nervous system in isolated mouse atria and human atrial appendages [53,54]. Additionally, studies have shown that *TNF-α* affects the expression of Cx43, which plays a crucial role in regulating the conductance of Cx43 channels [55,56]. Additional research should delve into the intricate relationship between *TNF-α*, symptomatic tone, Cx43 expression, and the development of AF in males with OSA.

Hypoxemia may induce cardiac remodeling by influencing the expression of HIF-1 and 2 [42]. The repetitive deoxygenation and reoxygenation characteristics of OSA resemble ischemia–reperfusion injury, elevating reactive oxygen species production, vascular inflammation, and blood pressure [43]. These processes collectively contribute to myocardial injury and may lead to the development of AF in OSA patients. In the current study, *HIF-1α* expression was significantly elevated in female OSA patients with and without AF compared to the control group. This finding suggests that hypoxia may play a role in the pathogenesis of OSA in women. However, this study could not demonstrate a distinction in *HIF-1α* expression between females with and without AF who had OSA.

The enrollment of a limited number of patients, especially females, is a potential limitation of our study. With only four female patients diagnosed with both OSA and AF in our study, we may be limited in demonstrating the possible mechanisms of AF development in OSA patients. A previous Taiwanese study found that the number of female patients with significant OSA was significantly lower than that of male patients with significant OSA (male/female: 2.1-fold difference) [32]. Additionally, it reported that the risk of incident AF in male OSA patients was higher than in female patients (HR: 1.9) [31]. The crude risk ratio in male patients with OSA and AF compared to female patients was approximately 4-fold higher. We observed a similar trend in this study, with 19 male patients and 4 female patients diagnosed with significant OSA and AF (19/4 = 4.8). That trend explains why we enrolled a limited number of female patients with OSA and AF in this study. The sex distribution of OSA patients with AF in our study was similar to that reported in previous studies. Although a previous study using the US nationwide health insurance database showed that there was little difference between the incidence of arrhythmia in women with OSA compared with matched male patients [57], a separate cohort study conducted in Taiwan also demonstrated a lower prevalence of OSA and AF burden among women [58]. These findings suggest potential variations or discrepancies among different racial groups. This substantial sex difference underscores the importance of conducting further studies specifically focused on females, which could yield valuable insights into the underlying mechanisms. Additionally, a larger case–control study should be performed to validate this study’s main finding.

## 5. Conclusions

In conclusion, our study revealed sex-specific differences in the risk factors and biomarkers associated with AF development in patients with OSA. Further studies should be conducted to investigate the distinct pathophysiological mechanisms in males and females.

## Figures and Tables

**Figure 1 biomedicines-12-01160-f001:**
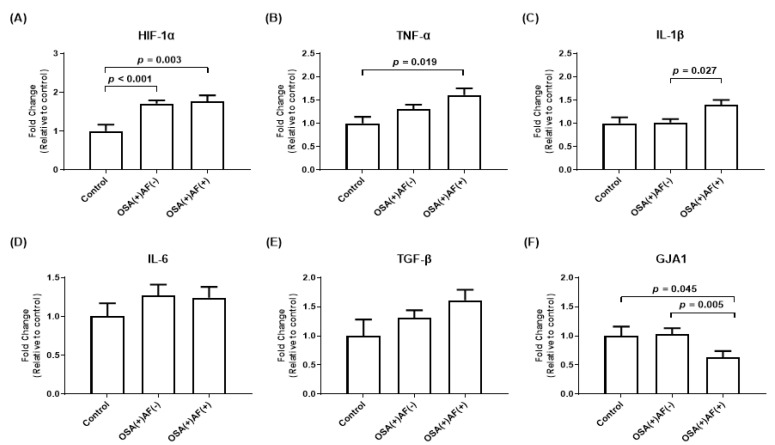
The mRNA gene expression of *HIF-1α* (**A**) and inflammatory cytokines (**B**–**F**) was measured in HL-1 cells that were treated with exosomes obtained from OSA patients with and without AF, as well as from control subjects. The *y*-axis represents the fold changes in mRNA expression levels in HL-1 cells incubated with exosomes obtained from those OSA patients with AF (n = 23) and without AF (n = 103) compared to control subjects (n = 28). GJA1, gap junction alpha-1; HIF, hypoxia-inducible factor; IL, interleukin; OSA, obstructive sleep apnea; TGF, transforming growth factor; TNF, tumor necrosis factor.

**Figure 2 biomedicines-12-01160-f002:**
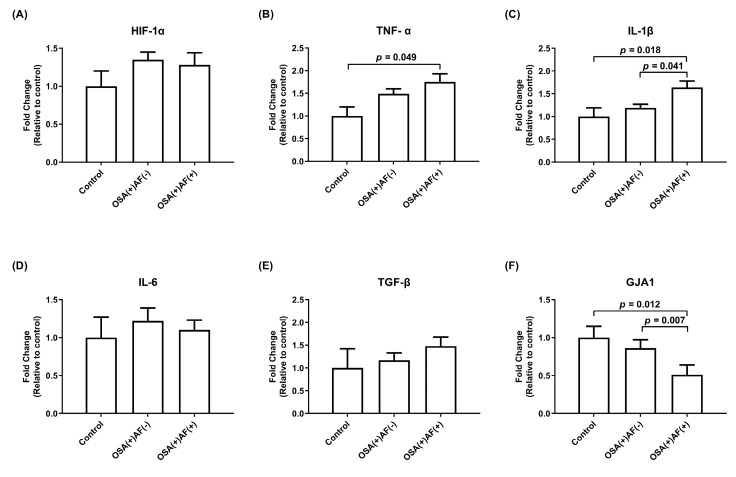
The mRNA gene expression of *HIF-1α* (**A**) and inflammatory cytokines (**B**–**F**) in HL-1 cells treated with exosomes obtained from male significant OSAS patients with and without AF, as well as control subjects. The *y*-axis represents the fold changes in mRNA expression levels in HL-1 cells incubated with exosomes obtained from male OSA patients with AF (n = 19) and without AF (n = 78) compared to control subjects (n = 13). GJA1, gap junction alpha-1; HIF, hypoxia-inducible factor; IL, interleukin; OSAS, obstructive sleep apnea syndrome; TGF, transforming growth factor; TNF, tumor necrosis factor.

**Figure 3 biomedicines-12-01160-f003:**
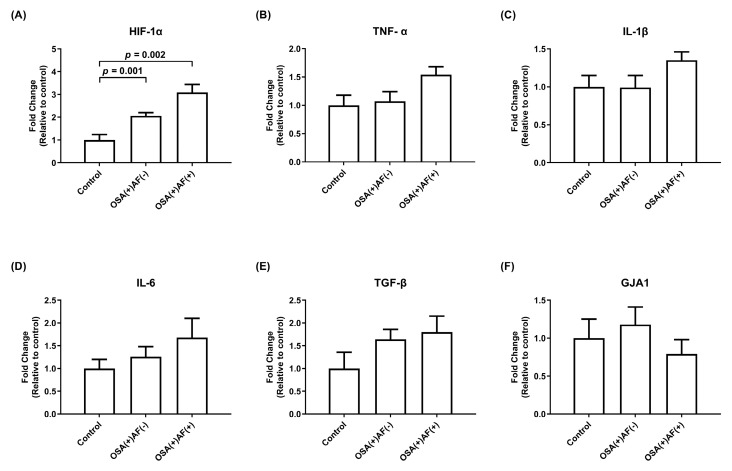
The mRNA gene expression of *HIF-1α* (**A**) and inflammatory cytokines (**B**–**F**) in HL-1 cells treated with exosomes obtained from significant female OSA patients with and without AF, as well as control subjects. The *y*-axis represents the fold changes in mRNA expression levels in HL-1 cells incubated with exosomes derived from female OSA patients with AF (n = 4) and without AF (n = 25) compared to control subjects (n = 15). GJA1, gap junction alpha-1; HIF, hypoxia-inducible factor; IL, interleukin; OSAS, obstructive sleep apnea syndrome; TGF, transforming growth factor; TNF, tumor necrosis factor.

**Table 1 biomedicines-12-01160-t001:** Baseline characteristics of OSA patients with and without AF and control subjects.

Variable	OSA with AF (*n* = 23)	OSA without AF (*n* = 103)	Control (*n* = 28)	*p*-Value
Age (years)	61.4 (9.7) ^b^	54.3 (10.8) ^a^	53.3 (9.9) ^a^	0.009
Male sex	19 (82.6) ^b^	78 (75.7) ^b^	13 (46.4) ^a^	0.004
BMI (kgs/m^2^)	27.0 (3.3) ^b^	26.3 (3.3) ^b^	24.4 (3.2) ^a^	0.010
Alcohol consumption	3 (13)	19 (18.4)	6 (21.4)	0.737
AF type				<0.001
Paroxysmal AF	13 (56.5)	0	0	
Persistent AF	10 (43.5)	0	0	
HTN	12 (52.2)	37 (35.9)	4 (25.0)	0.147
DM	5 (21.7)	8 (7.8)	3 (10.7)	0.160
HFpEF or HFmrEF	3 (13.0)	0	0	0.003
CAD	2 (8.7)	3 (2.9)	1 (3.6)	0.288
Stroke	2 (8.7)	4 (3.9)	0	0.213
Vital signs at enrollment				
Systolic blood pressure	135.9 ± 17.1	134.8 ± 14.4	133.4 ± 16.8	0.852
Diastolic blood pressure	79.1 ± 11.0	77.8 ± 10.0	75.5 ± 11.4	0.450
Heart rate	78.4 ± 11.9	80.9 ± 12.1	84.3 ± 14.5	0.237
Medication				
ACEI/ARB	9 (39.1)	30 (29.1)	5 (17.9)	0.241
Beta-blockers	14 (60.9)	25 (24.3)	8 (28.6)	0.003
Calcium channel blockers	13 (56.5%)	23 (22.3)	6 (21.4)	0.003
Diuretics	2 (8.7)	8 (7.8)	3 (10.7)	0.883
Oral antidiabetic drugs	3 (13)	7 (6.8)	3 (10.7)	0.555
Statins	5 (21.7)	35 (34)	8 (28.6)	0.491
Antiarrhythmic drugs	7 (30.4)	0	0	<0.001
Aspirin	2 (8.7)	13 (12.6)	4 (14.3)	0.824
P2Y12 inhibitors	1 (4.3)	7 (6.8)	1 (3.6)	0.769
Direct oral anticoagulants	12 (52.2)	0	0	<0.001
PSG data				
ESS	8.0 (4.0–11.0)	9.0 (6.0–12.0)	9.0 (7.0–12.0)	0.398
AHI_TST_ (/h)	39.5 (26.0–63.5) ^b^	40.3 (24.8–57.0) ^b^	8.0 (4.5–11.2) ^a^	<0.001
AHI_REM_ (/h)	44.9 (37.7–59.1) ^b^	48.2 (32.0–63.2) ^b^	13.5 (9.8–21.0) ^a^	<0.001
AHI_NREM_ (/h)	35.8 (23.3–63.9) ^b^	40.4 (21.9–56.9) ^b^	6.7 (3.6–9.4) ^a^	<0.001
ODI_TST_ (/h)	30.4 (14.4–54.1) ^b^	26.2 (14.2–44.3) ^b^	4.7 (1.4–7.5) ^a^	<0.001
ODI_REM_ (/h)	37.9 (23.2–55.0) ^b^	41.0 (18.4–55.3) ^b^	9.6 (5.2–17.9) ^a^	<0.001
ODI_NREM_ (/h)	31.6 (11.6–56.5) ^b^	23.3 (10.7–43.2) ^b^	3.9 (1.4–6.5) ^a^	<0.001
Lowest SpO_2_ (%)	79.0 (62.0–88.0) ^b^	80.0 (73.0–85.0) ^b^	86 (85–90.7) ^a^	<0.001
Mean SpO_2_ (%)	92.5 (91.5–94.6) ^b^	93.8 (92.2–94.9) ^b^	94.7 (93.3–95.9) ^a^	0.010
SpO_2_ < 90%	7.6 (4.2–22.0) ^b^	6.0 (2.2–16.7) ^b^	1.7 (0.2–7.0) ^a^	0.008
Echocardiography				
Aorta (mm)	33.3 (4.3)	32.5 (3.7)	31.6 (4.5)	0.337
LA (mm)	40.3 (6.0) ^b^	35.9 (4.4) ^a^	34.2 (5.7) ^a^	<0.001
IVS (mm)	11.8 (2.6)	11.5 (1.7)	11.0 (1.3)	0.337
LVPW (mm)	9.5 (1.9)	9.2 (1.5)	9.1 (1.3)	0.590
LVEDV (mL)	108.7 (41.3)	110.8 (27.6)	105.9 (23.0)	0.787
LVESV (mL)	42.2 (26.7)	34.5 (17.3)	29.8 (7.6)	0.072
LVEF (%)	63.9 (9.1) ^b^	69.2 (7.2) ^a^	71.4 (3.3) ^a^	0.001
E/e’	10.4 (3.3) ^b^	8.3 (2.6) ^a^	9.0 (4.2) ^ab^	0.036

The data are presented as means (standard deviations), medians (interquartile ranges), or numbers (percentages). Different letters (a and b) in the columns denote statistically significant differences (*p* < 0.05, Bonferroni). AF, atrial fibrillation; AHI, apnea–hypopnea index; BMI, body mass index; CAD, coronary artery disease; DM, diabetes mellitus; ESS, Epworth Sleepiness Scale; HFpEF or HFmrEF, heart failure with preserved ejection fraction or heart failure with mildly reduced ejection fraction; HTN, hypertension; IVS, interventricular septum; LA, left atrium; LVEDV, left ventricular end-diastolic volume; LVESV, left ventricular end-systolic volume; LVEF, left ventricular ejection fraction; LVPW, left ventricular posterior wall; ODI, oxygen desaturation index; OSA, obstructive sleep apnea; PSG, polysomnography; SpO_2_, oxygen saturation.

**Table 2 biomedicines-12-01160-t002:** Baseline characteristics of male OSA patients with and without AF and control subjects.

Male (110)				
Variable	OSA with AF (*n* = 19)	OSA without AF (*n* = 78)	Control (*n* = 13)	*p*-Value
Age (years)	60.5 (9.5) ^b^	51.8 (10.4) ^a^	50.0 (9.9) ^a^	0.003
BMI (kgs/m^2^)	26.7 (3.6) ^b^	26.6 (3.1) ^b^	23.6 (2.6) ^a^	0.005
Alcohol consumption	3 (15.8)	18 (23.1)	5 (38.5)	0.326
AF type				<0.001
Paroxysmal AF	11 (57.9)	0	0	
Persistent AF	8 (42.1)	0	0	
HTN	9 (47.4)	27 (34.6)	4 (30.8)	0.549
DM	3 (15.8)	5 (6.4)	1 (7.7)	0.335
HFpEF or HFmrEF	3 (15.8)	0	0	0.006
CAD	2 (10.5)	2 (2.6)	0	0.203
Stroke	2 (10.5)	4 (5.1)	0	0.400
Vital signs at enrollment				
Systolic blood pressure	134.1 ± 16.1	135.6 ± 13.7	134.2 ± 19.4	0.889
Diastolic blood pressure	78.8 ± 10.8	79.1 ± 9.9	78.7 ± 11.9	0.988
Heart rate	77.9 ± 11.0	82.5 ± 11.6	85.1 ± 15.0	0.200
Medication				
ACEI/ARB	7 (36.8)	20 (25.6)	3 (23.1)	0.578
Beta-blockers	12 (63.2)	14 (17.9)	3 (23.1)	<0.001
Calcium channel blockers	11 (57.9)	16 (20.5)	3 (23.1)	0.004
Diuretics	1 (5.3)	6 (7.7)	1 (7.7)	0.934
Oral antidiabetic drugs	2 (10.5)	6 (7.7)	2 (15.4)	0.652
Statins	4 (21.1)	21 (26.9)	2 (15.4)	0.621
Antiarrhythmic drugs	7 (36.8)	0	0	<0.001
Aspirin	2 (10.5)	11 (14.1)	2 (15.4)	0.903
P2Y12 inhibitors	0	4 (5.1)	0	0.427
Direct oral anticoagulants	7 (36.8)	0	0	<0.001
PSG data				
ESS	8.0 (4.0–11.0)	9.0 (6.0–12.0)	9.0 (5.0–11.0)	0.480
AHI_TST_ (/h)	39.5 (26.0–63.0) ^b^	42.3 (25.7–58.3) ^b^	7.6 (3.4–10.6) ^a^	<0.001
AHI_REM_ (/h)	45.0 (37.7–58.5) ^b^	48.0 (31.0–62.9) ^b^	13.5 (5.7–19.8) ^a^	<0.001
AHI_NREM_ (/h)	34.7 (23.3–63.9) ^b^	44.1 (23.2–58.4) ^b^	6.3 (3.2–8.7) ^a^	<0.001
ODI_TST_ (/h)	29.9 (13.4–62.0) ^b^	27.5 (13.0–45.1) ^b^	4.6 (1.1–6.8) ^a^	<0.001
ODI_REM_ (/h)	39.1 (22.7–55.9) ^b^	41.1 (18.9–55.0) ^b^	7.3 (1.7–12.8) ^a^	<0.001
ODI_NREM_ (/h)	31.2 (11.6–61.8) ^b^	29.3 (11.2–44.1) ^b^	4.5 (1.8–6.4) ^a^	<0.001
Lowest SpO_2_ (%)	78.0 (62.5–87.0) ^b^	79.0 (73.0–84.0) ^b^	89 (85–91.5) ^a^	<0.001
Mean SpO_2_ (%)	92.2 (91.0–94.9) ^b^	93.8 (92.3–95.0) ^ab^	94.6 (93.4–95.7) ^a^	0.035
SpO_2_ < 90%	11.6 (4.2–26.4)	5.9 (2.0–18.5)	1.30 (0.25–7.5)	0.054
Echocardiography				
Aorta (mm)	34.0 (4.1)	33.5 (3.5)	33.3 (5.2)	0.881
LA (mm)	40.0 (6.3) ^b^	36.0 (4.7) ^a^	36.4 (3.5) ^ab^	0.011
IVS (mm)	11.7 (2.8)	11.6 (1.8)	11.4 (0.8)	0.904
LVPW (mm)	9.6 (2.1)	9.3 (1.5)	8.7 (0.6)	0.446
LVEDV (mL)	112.4 (44.0)	114.5 (27.8)	119.6 (21.6)	0.851
LVESV (mL)	43.7 (28.6)	36.6 (19.0)	33.2 (8.0)	0.332
LVEF (%)	64.0 (7.8) ^b^	68.3 (7.8) ^ab^	72.3 (4.1) ^a^	0.027
E/e’	10.3 (3.5) ^b^	7.7 (2.1) ^a^	6.6 (2.2) ^a^	0.001

The data are presented as means (standard deviations), medians (interquartile ranges), or numbers (percentages). Different letters (a and b) in the columns denote statistically significant differences (*p* < 0.05, Bonferroni). AF, atrial fibrillation; AHI, apnea–hypopnea index; BMI, body mass index; CAD, coronary artery disease; DM, diabetes mellitus; ESS, Epworth Sleepiness Scale; HFpEF or HFmrEF, heart failure with preserved ejection fraction or heart failure with mildly reduced ejection fraction; HTN, hypertension; IVS, interventricular septum; LA, left atrium; LVEDV, left ventricular end-diastolic volume; LVESV, left ventricular end-systolic volume; LVEF, left ventricular ejection fraction; LVPW, left ventricular posterior wall; ODI, oxygen desaturation index; OSA, obstructive sleep apnea; PSG, polysomnography; SpO_2_, oxygen saturation.

**Table 3 biomedicines-12-01160-t003:** Baseline characteristics of female OSA patients with and without AF and control subjects.

Female (44)				
Variable	OSA with AF (*n* = 4)	OSA without (*n* = 25)	Control (*n* = 15)	*p*-Value
Age (years)	65.5 (11.1)	62.0 (7.9)	56.2 (9.2)	0.069
BMI (kgs/m^2^)	28.3 (0.9)	25.2 (3.8)	25.4 (3.6)	0.270
Alcohol consumption	0	1 (4)	1 (6.7)	0.834
AF type				<0.001
Paroxysmal AF	2 (50)	0	0	
Persistent AF	2 (50)	0	0	
HTN	3 (75.0)	10 (40.0)	3 (20.0)	0.092
DM	2 (50.0)	3 (12.0)	2 (13.3)	0.179
HFpEF or HFmrEF	0	0	0	-
CAD	0	1 (4.0)	1 (6.7)	1.000
Stroke	0	0	0	-
Vital signs at enrollment				
Systolic blood pressure	144.5 ± 21.9	132.2 ± 16.3	132.8 ± 14.6	0.376
Diastolic blood pressure	80.5 ± 13.3	74.1 ± 9.2	72.6 ± 10.5	0.386
Heart rate	80.5 ± 17.6	75.9 ± 12.6	83.6 ± 14.4	0.243
Medication				
ACEI/ARB	2 (50)	10 (40)	2 (13.3)	0.154
Beta-blockers	2 (50)	11 (44)	5 (33.3)	0.744
Calcium channel blockers	2 (50)	7 (28)	3 (20)	0.485
Diuretics	1 (25)	2 (8)	2 (13.3)	0.584
Oral antidiabetic drugs	1 (25)	1 (4)	1 (6.7)	0.302
Statins	1 (25)	14 (56)	6 (40)	0.392
Antiarrhythmic drugs	0	0	0	
Aspirin	0	2 (8)	2 (13.3)	0.683
P2Y12 inhibitors	1 (25)	3 (12)	1 (6.7)	0.584
Direct oral anticoagulants	1 (25)	0	0	0.006
PSG data				
ESS	7.5 (1.7–11.0)	9.0 (5.5–15.5)	9.0 (8.0–12.0)	0.541
AHI_TST_ (/h)	50.1 (23.5–64.9) ^b^	31.0 (22.2–47.4) ^b^	9.6 (5.8–11.8) ^a^	<0.001
AHI_REM_ (/h)	41.3 (38.4) ^ab^	51.8 (33.1–63.3) ^b^	14.4 (9.9–23.1) ^a^	<0.001
AHI_NREM_ (/h)	49.6 (21.1–64.9) ^b^	24.6 (21.3–44.4) ^b^	8.9 (4.7–9.6) ^a^	<0.001
ODI_TST_ (/h)	35.4 (18.4–49.5) ^b^	23.6 (14.7–34.0) ^b^	4.9 (1.7–8.6) ^a^	<0.001
ODI_REM_ (/h)	34.1 (25.3) ^ab^	40.6 (13.9–55.3)^b^	12.8 (6.8–22.9) ^a^	0.006
ODI_NREM_ (/h)	36.0 (16.6–49.2) ^b^	20.8 (10.3–30.3) ^b^	3.3 (1.0–6.8) ^a^	<0.001
Lowest SpO_2_ (%)	85.0 (79.0–88.0) ^b^	82.0 (74.2–86.7) ^ab^	86.0 (83.0–90.0) ^a^	0.021
Mean SpO_2_ (%)	93.3 (92.6–94.1)	93.7 (92.0–94.7)	95.0 (93.3–96.0)	0.249
SpO_2_ < 90%	5.2 (2.9–6.3)	6.0 (2.4–11.6)	2.1 (0.1–5.8)	0.162
Echocardiography				
Aorta (mm)	30.5 (4.7)	29.8 (2.7)	30.3 (3.7)	0.874
LA (mm)	42.0 (4.9) ^b^	35.6 (3.6) ^ab^	32.5 (6.5) ^a^	0.005
IVS (mm)	12.0 (1.4)	11.2 (1.4)	10.6 (1.4)	0.256
LVPW (mm)	9.2 (0.9)	8.9 (1.4)	9.3 (1.7)	0.759
LVEDV (mL)	91.0 (19.7)	101.0 (25.1)	95.6 (18.7)	0.636
LVESV (mL)	34.7 (15.6)	28.8 (10.0)	27.2 (6.5)	0.423
LVEF (%)	63.7 (7.8) ^b^	71.6 (4.7) ^a^	70.8 (2.6) ^a^	0.012
E/e’	10.9 (1.9)	9.9 (3.2)	10.2 (4.5)	0.906

The data are presented as means (standard deviations), medians (interquartile ranges), or numbers (percentages). Different letters (a and b) in the columns denote statistically significant differences (*p* < 0.05, Bonferroni). AF, atrial fibrillation; AHI, apnea–hypopnea index; BMI, body mass index; CAD, coronary artery disease; DM, diabetes mellitus; ESS, Epworth Sleepiness Scale; HFpEF or HFmrEF, heart failure with preserved ejection fraction or heart failure with mildly reduced ejection fraction; HTN, hypertension; IVS, interventricular septum; LA, left atrium; LVEDV, left ventricular end-diastolic volume; LVESV, left ventricular end-systolic volume; LVEF, left ventricular ejection fraction; LVPW, left ventricular posterior wall; ODI, oxygen desaturation index; OSA, obstructive sleep apnea; PSG, polysomnography; SpO_2_, oxygen saturation.

## Data Availability

The data underlying this article will be disclosed by the corresponding author upon reasonable request.

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
