# Peer review of "Sex Differences in the Expression of Cardiac Remodeling and Inflammatory Cytokines in Patients with Obstructive Sleep Apnea and Atrial Fibrillation"

_biomedicines, 2024, doi:10.3390/biomedicines12061160_

Round 1

Reviewer 1 Report (Previous Reviewer 1)

Comments and Suggestions for Authors

Dear Authors,

I have read the improved version of your manuscripts and am satisfied with your responses to my questions.  

I suggest you consider to perform a larger case-control study in the future to better address this interesting topic.

Thank you again for your efforts

Author Response

Dear reviewer, Thank you for your comment. To better address this interesting topic, we will perform a larger case-control study in the future. Thanks again.

Reviewer 2 Report (Previous Reviewer 2)

Comments and Suggestions for Authors

Author Chun-Ting Shih is to be commended for the well-executed presentation of the new version.

The authors have made notable efforts to address most of my concerns in this version.

However, some aspects of the revisions remain inadequate. Rectifying these issues is necessary for the paper to be considered for publication.

I had requested additional information regarding the treatment history for SAS and AF, but the provided explanation appears to be insufficient.

Specifically, details about CPAP or AF Ablation were not included in this revision.

Furthermore, there was no mention of sample size in the reviesd manuscript.

Failure to specify an appropriate sample size prior to conducting the study represents a critical flaw in the research design. If a sample size was not determined, an explanation for this omission is required. Alternatively, if a target sample size was set but not achieved, the reasons for this shortfall should be elucidated.

Author Response

  1. Thank you for your recommendation. As you stated, the treatment history could potentially impact the outcomes. Following enrollment, participants promptly underwent polysomnography (PSG), peripheral blood (PB) sampling, and echocardiography. In this investigation, patients did not undergo CPAP therapy or AF ablation until they completed all evaluations. So, we believe that the treatment will not influence the result. We have mentioned this information in the methods section of our revised manuscript. (Page 2, lines 82-84)
  2. Thank you for your recommendation. Our recent study published in Biomedicines (Biomedicines 2021, 9, doi:10.3390/biomedicines9101463.) found that OSA patients with AF had an LA size of 40.3 ± 6.4 mm. In contrast, OSA patients without AF had an LA size of 35.7 ± 4.6 mm. The patient-number ratio was calculated to be 1:4 when comparing individuals with and without AF. Based on a one-sided type I error rate of 0.025 and a power of 0.9, it was determined that a minimum of 100 OSA patients needed to be enrolled in the study. Out of these, 20 OSA patients had AF, and 80 did not have AF. Moreover, according to previous studies, the ratio of females to males was 1:4. The projected minimum number of female patients diagnosed with OSA and AF was 4, whereas the minimal number of male patients was 16. In this study, the number of OSA patients with AF was 23, consisting of 19 males and 4 females. The number of OSA patients without AF was 103. The number of enrolled individuals has already met our projected case numbers. We have mentioned this information in the methods section of our revised manuscript. (Page 4, lines 168-176)

Reviewer 3 Report (Previous Reviewer 3)

Comments and Suggestions for Authors

This reviewer considers that the authors well revised the paper, and has some comments as follows:

Minor comments:

1.       At the beginning of this paper, the authors should specify the type of article.

2.       In Figure 2D, the middle bar is labeled "OSAS," while the others are labeled "OSAS-AF." These labels should be consistent. Additionally, Figures 2 and 3 are relatively small and difficult to see; a presentation similar to Figure 1 would be preferable.

Author Response

  1. Thank you for your recommendation. We have added the “ Open Access” at the beginning of this paper.
  2. Thank you for your recommendation. We have relabeled Figure 2D and also replaced Figure 2 and Figure 3 with ones that are similar to Figure 1.

This manuscript is a resubmission of an earlier submission. The following is a list of the peer review reports and author responses from that submission.

Round 1

Reviewer 1 Report

Comments and Suggestions for Authors

Dear Authors,

The authors presented a single center prospective cohort of patients with sleep related breathing disorders (SRBD) and by the polysomnography they categorized the population in three groups: patients with obstructive sleep apnea (OSA) and concomitant atrial fibrillation (AF), patients with OSA and without AF and control group.

The main question of this research is analyze cardiac remodeling  and sex differences in expression of the inflammatory cytokines and GJA1 expression.

The main concern is about the total number of patients enrolled and the low number of female patients.

Like expected the patients with OSA and AF are older, with higher BMI and with a prevalence of male sex so the comparison is between 19 male and 4 female in this group, so it seems to me hard to assume sex-specific differences in biomarkers associated with AF development in patients with OSA.

Regarding cardiac remodeling the results are not original and probably affected by the bias in the composition of the population (older age and higher BMI are per se related to larger left atrium size and E/é ratio due to an higher prevalence of HFpEF in this population independently from AF presence).

Regarding the definition of the presence of HFpEF the authors didn’t mention about the determination of NT pro BNP in those patients.

In order to improve the manuscript probably is necessary to perform a case control study by matching the patients to a larger control population.

The abstract should be shorten by presenting only the main results (the other results should be presented in the Results section).

On Table 1 (Baseline characteristics of OSA patients with and without AF and control subjects) on  column 1 (Variable) Sex should be corrected in Male

Comments on the Quality of English Language

language used is not correct, there are conceptual errors and is not presentable.

Reviewer 2 Report

Comments and Suggestions for Authors

In the study by Chun-Ting Shih et al, they investigated “Sex Differences in the Expression of Cardiac Remodeling and Inflammatory Cytokines in Patients with Obstructive Sleep Apnea and Atrial Fibrillation.”

Obstructive sleep apnea (OSA) has long been recognized for its involvement in the onset of significant cardiovascular diseases such as atherosclerotic disorders, hypertension, atrial fibrillation, and heart failure, highlighting the importance of diagnosis and treatment. However, reports investigating its genetic expression has been scarce. This study was a highly intriguing investigation that examined the genetic expression of Gap junction alpha-1 and inflammatory cytokines by adding exosomes isolated from human-derived blood samples to HL-1 cells.

Concern #1

A significant limitation of this study was the small sample size, particularly with only four cases of female OSA patients with AF. Was a calculation performed to determine the statistically necessary sample size? Inadequate sample size is a significant limitation if not appropriately addressed.

Concern #2

Exosomes, being extremely small particles, require careful handling as they may be damaged or lost during processing or extraction. Although the experimental procedure was described, indicating a measure for replication, the limited quantity of exosomes obtainable from a single blood sample restricts the number of investigations feasible. Please provide an accurate protocol to address this concern.

Concern #3

OSA with AF patients in this study were older and had higher BMI compared to the control group, particularly pronounced in the male cohort. As age and BMI are implicated in AF onset, this bias remains unmitigated. Can you demonstrate that your results are not modulated by age or BMI, or provide a reasonable explanation? IF no, this is limitation of the study.

Concern #4

How was the control group selected? Please indicate the criteria for inclusion in the control group.

Concern #5

Important information regarding medication history and lifestyle factors (especially alcohol consumption), which are significant biases, was lacking. Additionally, it seemed that CAD and hyperlipidemia patients were also included. Please provide the rates of statin and aspirin usage, because of their anti-inflammatory effects.

Concern #6

There was limited information on patients’ characteristics. Did patients have histories of OSA or AF treatment? Additionally, was diagnosis wa paroxysmal AF or persistent AF? These biases could potentially influence the results.

Concern #7

Vital sign information (blood pressure and heart rate) is necessary as baseline data. Uncontrolled hypertension or AF tachycardia could biases the results.

Concern #8

The results showing the regression of gene expression in the GJA1 gene, known to encode connexins affecting cardiac conduction, in male OSA with AF patients are highly intriguing. The GJA1 gene has been associated with abnormal atrial automaticity and may be involved in both the onset and persistence of AF. Despite the significant discovery of the relationship between the GJA1 gene and OSA with AF in this study, the discussion regarding this finding is inadequate.

Reviewer 3 Report

Comments and Suggestions for Authors

This study aimed to explore sex differences in individuals with concurrent obstructive sleep apnea (OSA) and atrial fibrillation (AF), focusing on comorbidities, clinical presentation, cardiac remodeling, and pathophysiological mechanisms. The authors revealed sex-specific differences in the risk factors and biomarkers associated with AF development in patients with OSA. 

The reviewer considers that the authors well performed the present study, and has some comments as follows:

Major comments:

1.       In the Methods section, the authors are required to specify the hospitals or institutions from which patient cohorts were recruited. 

2.       Additionally, they should mention the recruitment of control subjects should be elucidated in the Methods section.

3.       It should be clarified whether patients who had previously treated for OSA were excluded from the study. 

4.       In Table 1, regarding the second variable, the term "sex" appears to be incorrectly used. Probably, it would be "male". 

5.       In the entire cohort or specifically the male participants, there was a significant age disparity observed. It raises the question of whether the mRNA expression levels were influenced by age. If age was a contributing factor to the expression levels, it would be prudent for the authors to conduct a comparative analysis of mRNA expressions across samples from age-matched groups. 

Minor comment:

6.       The authors have performed exosome extraction and utilized these in in vitro experiments. Are there any data whether analogous mRNA or protein expression profiles are observable in serum samples from patients with OSA/AF?